# Three-dimensional kinetic function of the lumbo-pelvic-hip complex during block start

**Natsuki Sado**[1,2]*, **Shinsuke Yoshioka**[2], **Senshi Fukashiro**[2]

**1** Faculty of Sport Sciences, Waseda University, Tokorozawa, Saitama, Japan, **2** Department of Life Sciences, The University of Tokyo, Tokyo, Japan

\* nsado@aoni.waseda.jp

**Data Availability Statement:** All relevant data are within the manuscript and its Supporting Information files.

## Abstract

Previous studies on joint kinetics during track and field block starts have been limited to lower-limb sagittal kinetics; however, we hypothesised that lumbopelvic extensors, lateral flexors, and hip abductors also act as substantial energy generators. The present study aimed to examine the three-dimensional lumbo-pelvic-hip kinetics to better understand the generation of mechanical energy during a block start. 3D kinematic and force data during block starts of 10 m maximal sprinting in 12 male sprinters (personal best in a 100 m sprint, 10.78 ± 0.19 s [range, 10.43–11.01 s]) were captured using a motion capture system and force platform. The three-dimensional lumbo-pelvic-hip kinetics were calculated. The peak lumbosacral extension torque (3.64 ± 0.39 Nm/kg) was significantly larger than any other lower-limb and lumbosacral torques (<3.0 Nm/kg). It was suggested that large lumbopelvic extension torques are needed during the block start to anchor the pelvis by cancelling out both hip extension torques acting on the pelvis, leading to hip extensor-induced thigh sagittal rotations rather than pelvic posterior tilt. During the double-stance phase, the lumbosacral extensors generated mechanical energy (0.35 ± 0.16 J/kg, 14 ± 4% of the sum of lumbosacral and lower-limb net joint work). During the single-stance phase, the sum of the net mechanical work by lumbosacral lateral flexors and front hip abductors was 0.35 ± 0.14 J/kg, which comprised 9 ± 3% of the sum of the net joint work. The results lead to the speculation of the importance of strengthening not only the leg extensors, but also the lumbopelvic extensors, lateral flexors, and hip abductors for block starts. Further training studies to verify this speculation will improve training strategies for the track and field block start performance.

## Introduction

In sprinting (60–400 m) races, sprinters have to start from the crouching position with starting blocks [1,2]. The starting performance accounted for 42% of the variance within the 100 m personal best (PB) in 154 male and female sprinters ranging 9.58–14.00 s [3]. Moreover, the velocity generated during a block start was related to the velocity at 40 m, which distinguished elite (PB: 9.95–10.29 s) from sub-elite sprinters (PB: 10.40–10.60 s) [4]. Thus, a better understanding of the kinetic demands in the block start would provide practical implications for performance improvement on overall sprinting races.

**Funding:** This work was supported by a Grant-in-Aid for JSPS Research Fellow grant number 16J08165 to NS.

**Competing interests:** The authors have declared that no competing interests exist.

Previous studies [5,6] have examined the power/work outputs of the lower-limb extensors during the start phase, demonstrating that the largest energy generators in both legs were the hip extensors and that the front knee extensors and bilateral ankle plantar-flexors also generated considerable amounts of kinetic energy. These work outputs were executed with strategies frequently used in a power-demanding task, such as the stretch-shortening cycle pattern (a power profile pattern of absorption followed by generation) and proximal-distal sequence [5,7].

Block starts involve not only the movements of the lower limbs, but also those of the trunk segments [8]. Previous studies [9,10] on vertical jumps, which are similar to block starts in that they involve large movements of trunk segments, confirmed that the angular displacement of the pelvis in the sagittal plane was smaller than that of the entire trunk (i.e., trunk extended). A simulation study [11] revealed that the sagittal rotations of the thorax and lumbar segment in a vertical jump were generated by the lumbopelvic extensors, not by the hip extensors, and that the jumping height in a squat jump performed without lumbopelvic extensors is approximately 15% lower than when performed with lumbopelvic extensors. Similar to a vertical jump [11], the lumbopelvic extensors might act as energy generators via the rotating trunk except the pelvis (i.e., extending trunk) during a block start. However, since there have been no studies on the lumbopelvic kinetics during block starts, the effects of the lumbopelvic extensors on energy generation remain unclear.

The lumbo-pelvic-hip complex can move in three-dimensions (3D). The modern human has well-developed muscles generating torques in frontal plane to adapt for bipedal locomotion [12]. Although kinetic studies on block starts [5,6] have only been examined in the sagittal plane, some previous studies [8,13] have additionally observed the lumbo-pelvic-hip 3D movements in block starts. Slawinski et al. [13] showed that the peak of the hip angular velocity does not reach the maximal angular velocity with just a flexion–extension, but with a combination of 3D movements. Debaere et al. [8] showed that the rear-leg side of the pelvis was elevated during a single stance phase (i.e., from rear-leg toe-off to front-leg toe-off) in block starts. The large hip and lumbopelvic frontal torque exertions have been confirmed in various single-leg stance movements [14–18]. Thus, the lumbosacral lateral flexors and hip abductors may also be important energy generators during a single stance phase of the block start. However, since 3D joint kinetics during block starts have not been investigated, the effects of 3D torque exertions on energy generation remain unclear.

Thus, a better understanding of the 3D lumbo-pelvic-hip kinetics might lead to enhancements in training strategies for the track and field block start. In order to expand the understanding of the biomechanical demand of executing block start with practical implication for performance improvement, we aimed to investigate the 3D lumbo-pelvic-hip kinetics during the block start. We hypothesized that the lumbosacral extensors, lateral flexors, and front hip abductors also act as substantial energy generators via rotating lumbar segment in the sagittal plane and elevating the rear-leg side of the pelvis, respectively.

## Materials and methods

### Participants

Twelve male sprinters participated in the study (mean ± SD: age, 21.7 ± 2.1 years; height, 1.76 ± 0.04 m; body mass, 66.0 ± 3.8 kg; PB in a 100 m sprint, 10.78 ± 0.19 s [range, 10.43–11.01 s]). They were free of injury for at least 6 months prior to participation. We performed a *post-hoc* power analysis for each comparison (see below) and confirmed the statistical power (1-$\beta$) of >0.99. They were members of the university's track and field team who trained regularly for sprinting events 4–5 days per week for >7 years. Prior to participation, they were

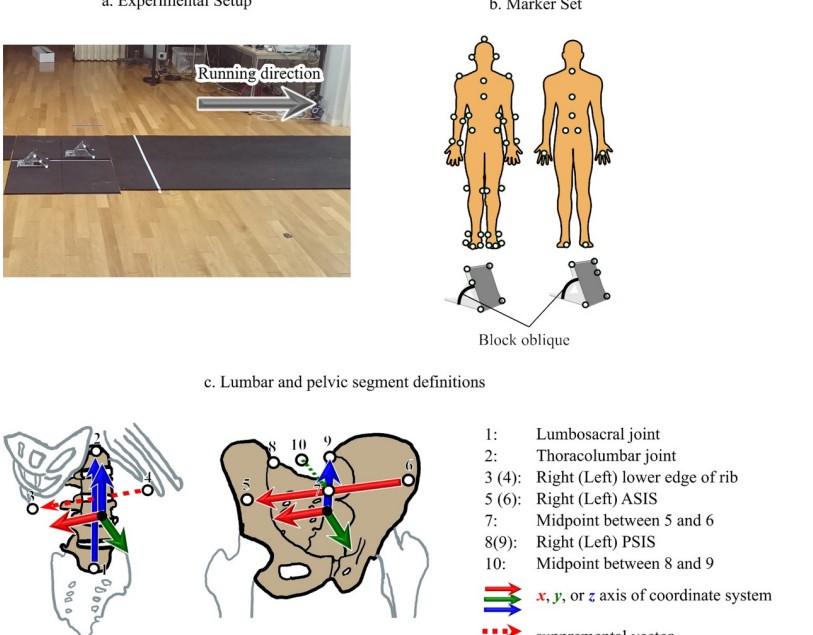

**Fig 1.** Experimental setup (a), marker setup (b), and definitions of lumbar and pelvic segments (c).

given a full explanation of the study and provided written informed consent. The Committee on Ethics of Experimental Research on Human Subjects, Graduate School of Arts and Sciences, The University of Tokyo approved the study protocol (reference number: 356–2).

## Experimental protocol

After a 30 min individualized warm-up, including some practice for the block start, each participant performed three 10 m sprints from the block start with a gun-like signal from a starter-signal tool (NT7713C, Nishi, Japan). The participants were instructed to run as fast as possible. Adequate recovery time (>3 min) was provided between trials to avoid fatigue.

The separated (i.e., not connected by a common bar) starting blocks (NF196BR and NF196BL, Nishi, Japan) were secured onto separate force plates (Fig 1A) using double-sided tapes (NW-N20, Nichiban, Japan) [19]. Using marker coordinates attached to the starting blocks, we confirmed that the antero-posterior movement of the centre of each block from onset of the movement to toe-off was little (< 1 mm). The anteroposterior distance between the blocks was set to each athlete's preferred distance. The mediolateral width of the blocks (approximately 25 cm) was consistent with that of a general starting block. The participants were able to change the oblique of the blocks.

All participants wore well-fitted clothing and their own running shoes with spikes. Reflective markers that were 20 mm in diameter were secured to each location of the body [16] and on the starting blocks (Fig 1B). A 14-camera motion capture system (Motion Analysis Corporation, USA) recorded the 3D coordinates of the positions of the reflective markers (sampling rate, 200 Hz). The $x$-, $y$-, and $z$-axes of the global coordinate system (GCS) defined the medial-lateral, anterior-posterior, and superior-inferior directions, respectively. The reaction force on each foot was measured using separate force platforms (Force Plate 9281E, Kistler, Switzerland) at a sampling rate of 2000 Hz and synchronized with the motion data.

## Data analysis

The following data processing was performed using MATLAB 2014a (MathWorks Inc., USA). Position coordinates of the markers were smoothened using a 4[th] order Butterworth, low-pass, digital filter with a cut-off frequency of 14 Hz based on a residual analysis [20]. The GRF data were also smoothed using the same Butterworth low-pass digital filter with the same cut-off frequency [21,22]. We analysed data from the onset of the movement to the front leg toe-off. The instant of the movement onset was defined as the frame when the vertical reaction force remained greater than 2 SD of that during the "set" position longer than 20 frames (0.1 s). The instant of each toe-off was defined as less than 10 N of the vertical reaction force on each foot. The block phase was then divided into two (double-stance and single-stance) phases by the instant of the toe-off of the rear leg [2].

The whole-body model and joint centre definition were consistent with our previous study [16]. The definitions of lumbar and pelvic segments were shown in Fig 1C. The CoM of the whole body and each segment inertia parameter were estimated using the anthropometric data and scaling equations in Dumas et al. [23,24].

To quantify the starting performance, the normalised averaged horizontal power (NAHP) [25] was calculated as:

$$NAHP = \frac{m\,v_{\mathrm{hori}}^{2}}{2\,\Delta t} \cdot \frac{1}{m\,g^{3/2}\,l^{1/2}} \tag{1}$$

where $m$ is the body mass, $v_{\mathrm{hori}}$ is the horizontal velocity at the end of block phase, $\Delta t$ is the duration of the block phase, $g$ is the gravitational acceleration, and $l$ is the leg length of the sprinter, respectively.

Right-handed local segment coordinate systems (SCSs) and joint coordinate systems (JCSs) were defined in each frame. For the lumbar SCS definition, the $z_{\mathbf{lumbar}}$ runs from the lumbosacral joint to the thoracolumbar joint, the $y_{\mathbf{lumbar}}$ is the cross product of the $z_{\mathbf{lumbar}}$ and the vector running from the left lower edge of rib to the right lower edge of rib, and the $x_{\mathbf{lumbar}}$ is the cross product of the $y_{\mathbf{lumbar}}$ and the $z_{\mathbf{lumbar}}$. For the pelvic SCS definition, the $x_{\mathbf{pelvis}}$ runs from the left ASIS to the right ASIS, the $z_{\mathbf{pelvis}}$ is the cross product of the $x_{\mathbf{pelvis}}$ and the vector running from the midpoint of PSISs to that of ASIS, and the $y_{\mathbf{pelvis}}$ is the cross product of the $z_{\mathbf{pelvis}}$ and the $x_{\mathbf{pelvis}}$. The details of the definitions in lower-limb segment SCSs can be found elsewhere [26]. The lumbar and pelvic 3D angles were calculated as the Cardan ($xyz$ sequence) angles of their SCSs relative to the GCS. The 3D joint angles were calculated using the JCS conventions.

Newton-Euler equations [20] were used to calculate the 3D joint torques with the transformation into the JCSs [27]. The position of the centre of pressure (CoP) on each block plane (Fig 2) was calculated as the point with respect to the component of the free-moment vector around the axes parallel to the block plane that was 0 [28] as shown below:

$$\left(r^{\mathbf{O}\rightarrow\mathbf{B}} + R_{\mathbf{BCS}\rightarrow\mathbf{GCS}}\,r^{\mathbf{B}\rightarrow\mathbf{CoP}'}\right) \times f + R_{\mathbf{BCS}\rightarrow\mathbf{GCS}} \begin{bmatrix} 0 \\ 0 \\ n_{z_{\mathrm{BCS}}} \end{bmatrix} = n^{\mathbf{total}} \tag{2}$$

where $r^{\mathbf{O}\rightarrow\mathbf{B}}$ is the position vector from the GCS origin to the block coordinate system (BCS) origin expressed in GCS, $R_{\mathbf{GCS}\rightarrow\mathbf{BCS}}$ is the transformation matrix from GCS to BCS, $r^{\mathbf{B}\rightarrow\mathbf{CoP}'}$ is the position vector from the BCS origin to the CoP expressed in BCS, $f$ is the GRF vector expressed in GCS, $\begin{bmatrix} 0 \\ 0 \\ n_{z_{\mathrm{BCS}}} \end{bmatrix}$ is the free-moment vector applied on the block plane ($x_{\mathbf{BCS}}\,y_{\mathbf{BCS}}$ vector

a. Definition of CoP

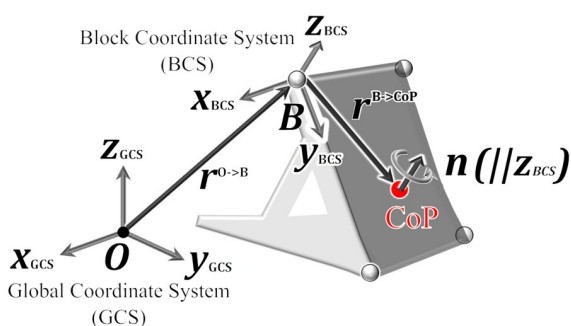

b. Example of trajectories of CoP

**Fig 2.** Calculation of the centre of pressure (CoP) (a), and a representative example of CoP trajectories (b).

plane) expressed in BCS, and $n^{\text{total}}$ is the applied moment vector around the origin of the GCS expressed in the GCS, respectively. From Eq 2, the position of the CoP ($r^{\text{O}\rightarrow\text{CoP}}$) can be calculated as:

$$r^{\text{O}\rightarrow\text{CoP}} = r^{\text{O}\rightarrow\text{B}} + R_{\text{BCS}\rightarrow\text{GCS}}r^{\text{B}\rightarrow\text{CoP}'} \tag{3}$$

$$r^{\text{B}\rightarrow\text{CoP}'} = \begin{bmatrix} -\dfrac{n_{y_{\text{BCS}}}^{\text{total}} - [r^{\text{O}\rightarrow\text{B}} \times f]_{y_{\text{BCS}}}}{f_{z_{\text{BCS}}}} \\[4mm] \dfrac{n_{x_{\text{BCS}}}^{\text{total}} - [r^{\text{O}\rightarrow\text{B}} \times f]_{x_{\text{BCS}}}}{f_{z_{\text{BCS}}}} \\[4mm] 0 \end{bmatrix} \tag{4}$$

where $n_{x_{\text{BCS}}}^{\text{total}}$ $n_{y_{\text{BCS}}}^{\text{total}}$ are the $x_{\text{BCS}}$ and $y_{\text{BCS}}$ components of the $n^{\text{total}}$ expressed in BCS, $[r^{\text{O}\rightarrow\text{B}} \times f]_{x_{\text{BCS}}}$ and $[r^{\text{O}\rightarrow\text{B}} \times f]_{y_{\text{BCS}}}$ are the $x_{\text{BCS}}$ and $y_{\text{BCS}}$ components of the $[r^{\text{O}\rightarrow\text{B}} \times f]$ expressed in BCS, and $f_{z_{\text{BCS}}}$ is the $z_{\text{BCS}}$ component of the GRF vector expressed in BCS, respectively.

The joint torque power was calculated using joint torque and angular velocity [26]. The net mechanical work was calculated as the numerical integration of the joint torque power.

Relative work was calculated as the percentage of each net mechanical work to the sum of the net mechanical work of the lumbosacral and bilateral lower-limb joints.

The mean value of the three trials was used as the representative value for each participant. Data normality was analysed using the Shapiro-Wilk test. After data normality was confirmed (Shapiro-Wilk $p>0.05$), a paired two-tailed $t$-test was used to compare the peak values between the lumbosacral extension torque and other lower limb and lumbosacral torques. To control the family-wise error rate, the alpha level of each $t$-test was adjusted with the Bonferroni method. Overall statistical significance was set at $\alpha<0.05$. The effect size of each comparison was determined as Cohen's $d$ as:

$$d = \frac{|\bar{x_1} - \bar{x_2}|}{\sigma_{\text{pooled}}} \tag{5}$$

where $\bar{x_1}$ and $\bar{x_2}$ are the mean values and $\sigma_{\text{pooled}}$ is the pooled SD. According to Cohen [29], the effect sizes can be classified as small ($\leq 0.49$), medium ($0.50$–$0.79$), and large ($\geq 0.80$). To assess the relationships between some lumbo-pelvic-hip kinetic variables and NAHP, we calculated two-tailed Pearson product–moment correlations ($r$). Statistical analyses were performed using SPSS 23 for Windows (SPSS Inc., Chicago, IL, USA).

## Results

The horizontal and vertical velocities at the end of the block phase were $3.31 \pm 0.13$ and $0.58 \pm 0.08$ m/s, respectively. The duration of the block phase was $0.36 \pm 0.03$ s. The relative duration of the double-stance phase ($0.18 \pm 0.02$ s) was $49.7 \pm 5.1\%$ of the overall phase. The NAHP was $0.55 \pm 0.05$.

The posterior (+) / anterior (-) tilt angles at the beginning and end of the block phase were $-94.8 \pm 10.3°$ and $-51.0 \pm 9.1°$, respectively, for the lumbar, and $-80.7 \pm 11.2°$ and $-52.0 \pm 10.5°$, respectively, for the pelvis, with angular displacements of $43.8 \pm 8.9°$ for the lumbar and $28.8 \pm 7.2°$ for the pelvis (Fig 3A and 3D). In the frontal plane, the elevation (+) / drop (-) angles of the rear-leg side of the pelvis (Fig 3E) at the beginning and end of the block start were $-4.9 \pm 3.2°$ and $11.3 \pm 3.5°$, respectively, with a pelvic elevation angular displacement of $16.2 \pm 3.2°$.

The lumbosacral extension torque was exerted during almost the entire block phase (Fig 4G) and reached a peak in the double-stance phase. The peak lumbosacral extension torque ($3.64 \pm 0.39$ Nm/kg) was significantly larger ($p<0.05$, Cohen's $d = 2.02$–$11.09$) than any other lower-limb and lumbosacral torques ($<3.0$ Nm/kg) (Fig 5). The lumbosacral extensors exerted a positive power during almost all block phases (Fig 4J red). Front hip exerted extension torque (Fig 4H), which exerted positive powers (Fig 4K). Rear hip extension (Fig 4I) torque was exerted during the double-stance phase, which exerted positive powers (Fig 4L). During the single-stance phase, rear hip flexors exerted positive power (Fig 4L). In the frontal plane, lumbosacral rear-leg side flexion torque (Fig 4G) and front hip abduction torque (Fig 4H) exerted positive powers during the single-stance phase (Fig 4J and 4K).

During the double-stance phase (Fig 6A), the main net energy generators ($>10\%$ of the sum of lumbosacral and lower-limb net joint work) were the lumbosacral extensors ($0.35 \pm 0.16$ J/kg, $14 \pm 4\%$ of total net generated energy), front hip extensors ($1.12 \pm 0.13$ J/kg, $45 \pm 5\%$), rear hip extensors ($0.57 \pm 0.28$ J/kg, $22 \pm 8\%$), and rear ankle plantar-flexors ($0.38 \pm 0.14$ J/kg, $15 \pm 5\%$). Furthermore, the main net energy generators during the single-stance phase (Fig 6B) were the front hip extensors ($1.30 \pm 0.29$ J/kg, $31 \pm 8\%$), front knee extensors ($0.79 \pm 0.22$ J/kg, $21 \pm 6\%$), and front ankle plantar-flexors ($1.15 \pm 0.19$ J/kg, $30 \pm 3\%$). In addition, the sum of the net generated energy by the lumbosacral rear-leg side flexors ($0.24 \pm 0.17$ J/kg, $6 \pm 4\%$) and front hip abductors ($0.11 \pm 0.11$ J/kg, $3 \pm 3\%$) was $9 \pm 3\%$ ($0.35 \pm 0.14$ J/kg).

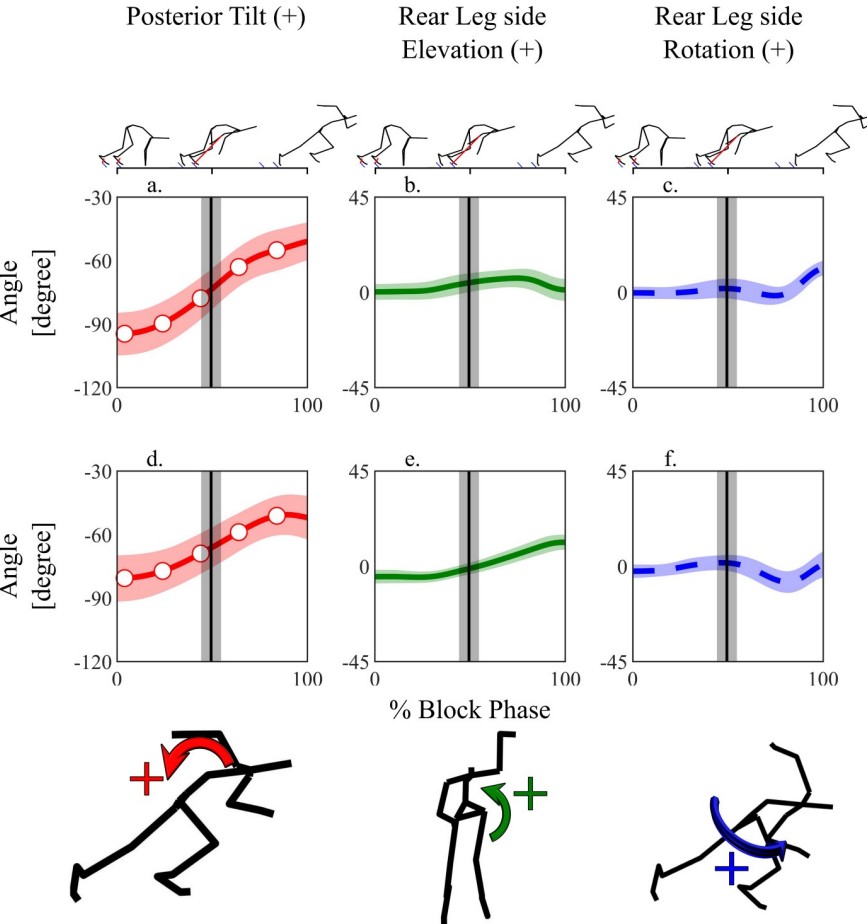

**Fig 3. Ensemble averages of the lumbar and pelvic Cardan angles relative to global coordinate system during the start phase.** Vertical lines show the instant of the rear foot toe-off.

The peak lumbosacral extension torque ($r = 0.67$, $p<0.05$, Fig 7A), the net mechanical work exerted by the lumbosacral extensors during the double-stance phase ($r = 0.63$, $p<0.05$, Fig 7B), and the sum of the net mechanical work exerted by the lumbosacral rear-leg side flexors and front hip abductors during single-leg stance phase ($r = 0.68$, $p<0.05$, Fig 7C) were positively correlated with the NAHP.

## Discussion

We aimed to investigate the 3D lumbo-pelvic-hip kinetics in block start. The main findings were as follows: (1) the peak lumbosacral extension torque was larger than any other peak torques, (2) the lumbosacral extensors generated mechanical energy 14 ± 4% of the sum of lumbosacral and lower-limb net joint work during double-stance phase and (3) the sum of generated energies by lumbosacral lateral flexors and front hip abductors comprised 9 ± 3% of the sum of the net joint work. To the knowledge of the authors, this is the first study to describe the kinetic roles of the lumbo-pelvic-hip complex during the track and field sprint start.

We found that the posterior pelvic tilt displacement was smaller than that of the lumbar, resulting in lumbosacral extension, similar to the vertical jump in previous studies [9,10]. Blache and Monteil [11] showed that the vertical jump height decreased approximately 15%

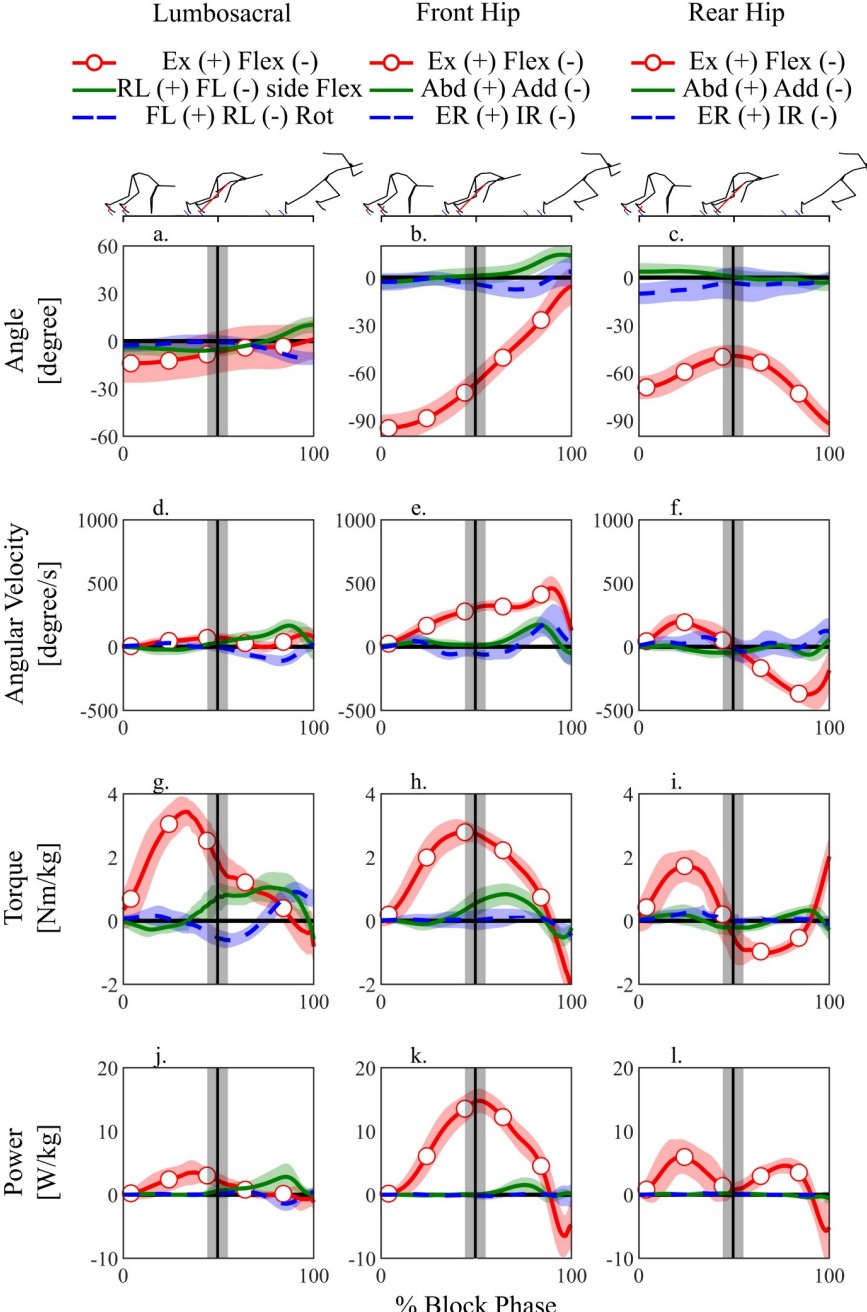

**Fig 4. Ensemble averages of the three-dimensional joint angles, angular velocities, torques, and powers at the lumbosacral and both hip joints during the start phase.** Vertical lines show the instant of the rear foot toe-off.

when the lumbopelvic extensor muscles were not considered in the simulation model. The lumbosacral extensors generated considerable amounts of kinetic energy (14% of the sum of lumbosacral and lower-limb net joint work during the double-stance phase), thus supporting our hypothesis. Further, we found the positive correlation between the net mechanical work

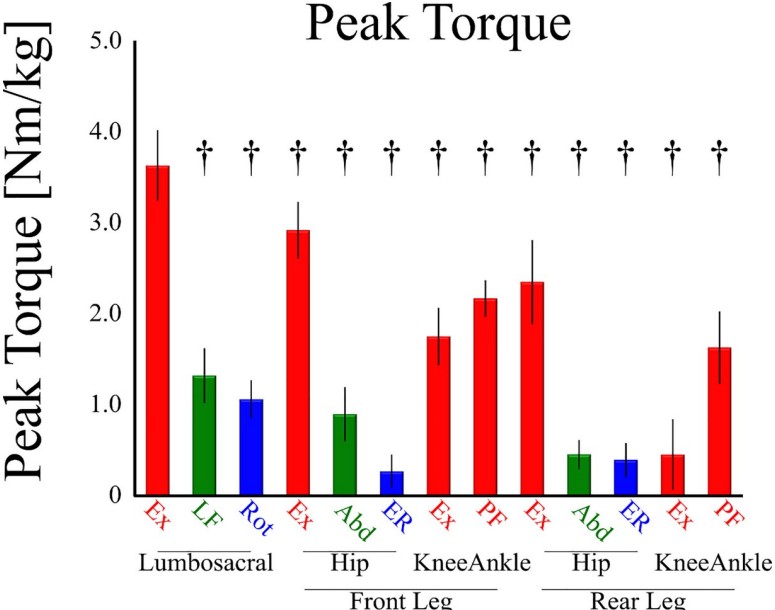

Ex: Extension, LF: Lateral Flexion, Rot: Rotation,
Abd: Abduction, ER: External Rotation, PF: Plantar Flexion
† : < Lumbosacral Extension torque ($p < 0.004$)

**Fig 5. Mean ± standard deviation of the peak torque during block phase.**

exerted by lumbosacral extensors and the starting performance. We speculate that, in a block start with the body leaning anteriorly, lumbopelvic extensors might contribute not only to body lift, but also to body propulsion by generating mechanical energy via rotating trunk segments except the pelvis in the sagittal plane (i.e., extending trunk).

The peak lumbosacral extension torque was significantly larger than any other lumbosacral and lower-limb torques, and positively correlated with the starting performance. This peak value appeared in the double-stance phase where both hip joints exerted extension torques. Anatomically, a hip extension involves a pelvic posterior tilt, whereas a lumbosacral extension involves a pelvic anterior tilt. Thus, the lumbosacral extension torque acted to anchor the pelvis in the sagittal plane by cancelling out hip extension torques acting on the pelvis, thereby indicating that both hip extensions induce thigh forward sagittal rotations (moving knee

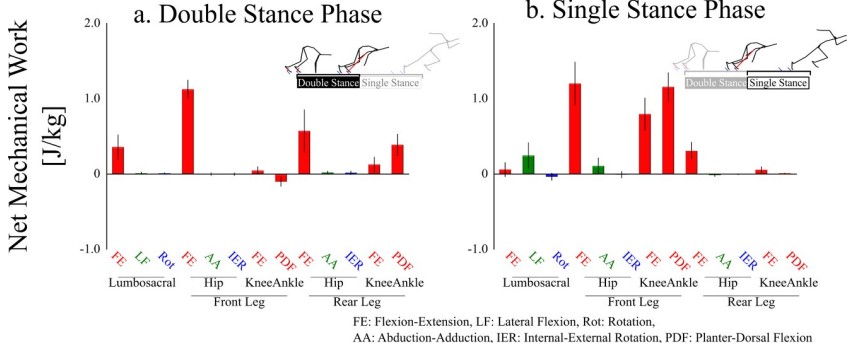

FE: Flexion-Extension, LF: Lateral Flexion, Rot: Rotation,
AA: Abduction-Adduction, IER: Internal-External Rotation, PDF: Planter-Dorsal Flexion

**Fig 6.** Mean ± standard deviation of net mechanical work during double-stance (a) and single-stance (b) phases.

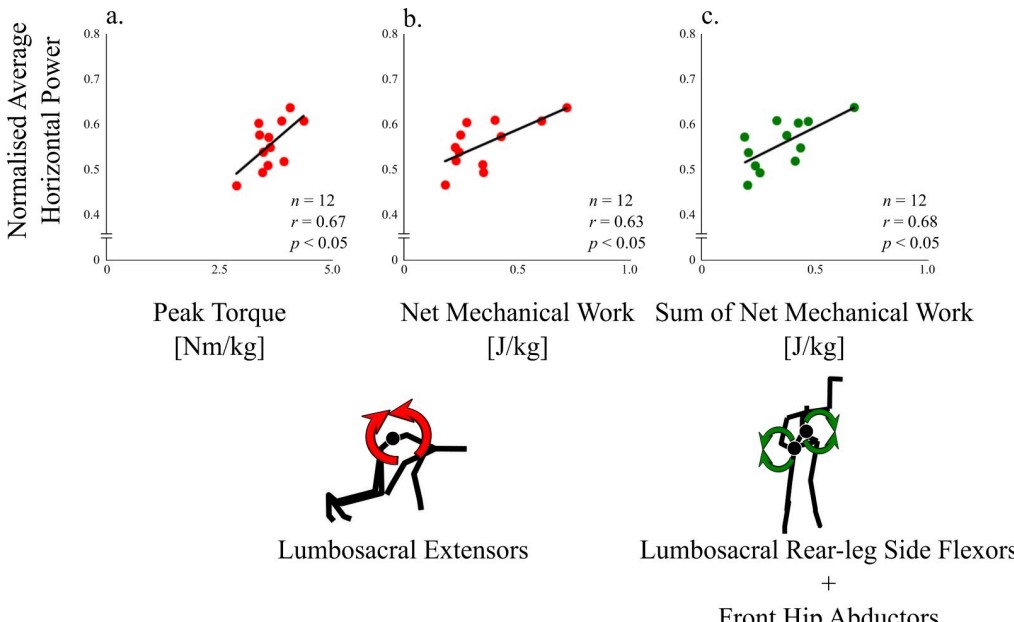

**Fig 7.** Relationships between lumbo-pelvic-hip variables [peak lumbosacral extension torque (a), net mechanical work exerted by lumbosacral extensors during double-stance phase (b), and sum of net mechanical work exerted by lumbosacral rear-leg side flexors and by front hip abductors during single-stance phase (c)] and starting performance (Normalised average horizontal power: NAHP).

backward relative to hip) rather than the pelvic posterior tilt. The thigh sagittal rotation is an important factor in the generation of horizontal velocity [30,31]. The relationship between the start performance and both hip extension movements has been shown from the kinematic [32] and kinetic [33] points of view. Thus, we indicated that lumbosacral extensors would contribute to perform block start not only by generating kinetic energy but also by leading to hip extensor-induced thigh sagittal rotations.

The rear-leg side of the pelvis was elevated during the single-stance phase, similar to Debaere et al. [8]. The front hip abductors and lumbosacral lateral flexors toward the rear-leg side, actions to elevate the pelvic rear-leg side, generated kinetic energy during the single-stance phase. The sum of the net mechanical work generated by the hip abductors and lumbosacral lateral flexors was approximately 10% of the total generated kinetic energy in the single-stance phase, suggesting that the frontal torque exertions positively contributed to the energy generation by elevating the pelvic rear-leg side.

The sum of the net mechanical work exerted by the front hip abductors and the lumbosacral rear-leg side flexors was positively correlated with the sprinting performance. The pelvis is in a considerably anteriorly tilted position during the block start. The elevation of a certain side of the pelvis in an upright posture would move the ipsilateral leg upward, resulting in a higher CoM [16]; however, elevation of the pelvis can contribute to not only body lift, but also forward propulsion of the CoM in an anteriorly tilted position. Thus, we speculated that hip abductors and lumbar lateral flexors contribute not only to body lift, but also to propulsion by elevating the rear-leg side of the pelvis. A mechanical work analysis alone cannot be used to quantify the effect of each movement on each propulsion and body lift. This point is a limitation of the present study and important future theme.

Our results suggested the importance to strengthen lumbopelvic extensor, lateral flexor and hip abductor muscles for performing block starts. The power profiles provide practical

suggestions for training strategies. In a block start, the lumbosacral extensors, lateral flexors, and hip abductors exerted mostly positive power, preceded by neglect able negative power generation, suggesting only a small countermovement involved in their action. Thus, the addition of exercises for pure concentric torque exertion skills or abilities to daily training programs might improve block start performance. Further longitudinal studies may be needed to examine the effect of exercises on their pure concentric exertion abilities on block start performance.

In conclusion, we found that lumbosacral extension torque was larger than other lumbosacral and lower-limb torques. It was suggested that the lumbopelvic extensors anchor the pelvis by cancelling out both hip extension torques acting on the pelvis, leading to hip extensor-induced thigh sagittal rotations rather than pelvic posterior tilt during a block start. The net mechanical work generated by the lumbosacral extensors during the double-stance phase ($14 \pm 4\%$) and the sum of the net mechanical work by the lumbopelvic rear-leg side flexors and front hip abductors during the single-stance phase ($9 \pm 3\%$) were also appreciable. The findings suggest the importance of their power capacities for block start performance. The power profiles and absence of a stretch-shortening cycle pattern imply that the training exercises to improve pure concentric torque exertion skills may be effective.

## Supporting information

**S1 File.**
(M)

**S2 File.**
(M)

**S3 File.**
(M)

## Acknowledgments

We would like to thank Akihiro Fujita for his help in recruiting the participants and Editage (www.editage.jp) for English language editing.

## Author Contributions

**Conceptualization:** Natsuki Sado.

**Data curation:** Natsuki Sado.

**Formal analysis:** Natsuki Sado.

**Funding acquisition:** Natsuki Sado.

**Investigation:** Natsuki Sado.

**Methodology:** Natsuki Sado.

**Project administration:** Natsuki Sado.

**Resources:** Natsuki Sado.

**Software:** Natsuki Sado.

**Supervision:** Shinsuke Yoshioka, Senshi Fukashiro.

**Validation:** Natsuki Sado.

**Visualization:** Natsuki Sado.

**Writing – original draft:** Natsuki Sado.

**Writing – review & editing:** Natsuki Sado, Shinsuke Yoshioka, Senshi Fukashiro.

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
