## [Decision Letter · Decision Letter 0]

23 Jan 2020

PONE-D-19-35792

Three-dimensional kinetic function of the lumbo-pelvic-hip complex during block start

PLOS ONE

Dear Dr. Sado,

Thank you for submitting your manuscript to PLOS ONE. After careful consideration, we feel that it has merit but does not fully meet PLOS ONE’s publication criteria as it currently stands. Therefore, we invite you to submit a revised version of the manuscript that addresses the points raised during the review process.

In your revision, please respond to all of the comments posed by the reviewers. Please specifically address Reviewer 2's request for more information/detail about the analyses used and consider adding mechanical power as a metric for comparison with previous studies.

We would appreciate receiving your revised manuscript within 60 days. To enhance the reproducibility of your results, we recommend that if applicable you deposit your laboratory protocols in protocols.io, where a protocol can be assigned its own identifier (DOI) such that it can be cited independently in the future. For instructions see: http://journals.plos.org/plosone/s/submission-guidelines#loc-laboratory-protocols

We look forward to receiving your revised manuscript.

Kind regards,

Alena Grabowski

Academic Editor

PLOS ONE

Journal Requirements:

2. We note that in your methods section you state that the runners selected for the study were all recovering from injuries. Please revise the manuscript to provide details of the injuries, or an explanation why this information is not relevant.

Additionally, PLOS ONE specifies that experiments, statistics, and other analyses are performed to a high technical standard; sample sizes are large enough to produce robust results; and methods are described in sufficient detail to allow another researcher to reproduce the experiment (http://journals.plos.org/plosone/s/criteria-for-publication#loc-3). We note that your data was processed using Matlab. Please provide the full code/scripts used to process the data or a link to a code repository where it can be accessed.

Reviewers' comments:

Reviewer's Responses to Questions

**Comments to the Author**

1. Is the manuscript technically sound, and do the data support the conclusions?

Reviewer #1: Yes

Reviewer #2: Yes

2. Has the statistical analysis been performed appropriately and rigorously? 

Reviewer #1: Yes

Reviewer #2: Yes

3. Have the authors made all data underlying the findings in their manuscript fully available?

Reviewer #1: Yes

Reviewer #2: No

4. Is the manuscript presented in an intelligible fashion and written in standard English?

Reviewer #1: Yes

Reviewer #2: Yes

5. Review Comments to the Author

Reviewer #1: The research question is new and the experimnts and analysis are very scientific. Also the manuscript is well written. Some minor comments:

* page 7 ...prefer to write local university ..not our university

* page 7 not a go signal ...use e.g. a gun signal

* page 14: not of 14 % of ..prefer 14 % of...

* page 17 ..in conclusion the first sentence is very long. S Make two short sentences.

Reviewer #2: Ms. Number.: PONE-D-19-35792

Title: Three-dimensional kinetic function of the lumbo-pelvic-hip complex during block start

Overview and general recommendation:

This study presents an analysis of the kinematics and kinetics of the lumbo-pelvic-hip complex and the lower extremity joints during the track and field sprint start. It is the first study that addresses the kinetics of this complex in the literature and therefore adds important knowledge to the field of Sports Biomechanics. The authors need to be commended for their efforts.

The data collection and analysis seems to be well performed. However, the authors should describe certain aspects of their analysis in greater detail. Further, it would be nice to add the calculation of the normalized average horizontal block power values for the analyzed start to allow for a comparison of the start performances in this study to the published literature. The manuscript would also benefit from several improvements in the quality of the English language (see my specific comments).

Specific comments:

Abstract:

Line 2: Please change to: “joint kinetics during track and field block starts”

Line 6: Please change to: “mechanical energy during a block start”

Line 6: Please change to: “start. 3D kinematic and force data”

Line 7: Please add the performance level (100 m PB times) here.

Line 8: Please change to: “using a motion capture”

Line 9: Please change to: “kinetics were calculated”

Line 11: Please change to: “during the block start”

Line 18-19: Please rephrase this statement. You have not analyzed the effects of strengthening these muscle groups on start performance in this study. Please clearly highlight that this statement is speculative at the moment and that further studies are needed to verify this hypothesis.

Introduction:

Line 22: A reference to the recent review of Bezodis et al. might be relevant here, since it provides a good overview on the current sprint start literature and also provides a phase classification for the sprint start. (Bezodis, N. E., Willwacher, S., & Salo, A. I. T. (2019). The Biomechanics of the Track and Field Sprint Start: A Narrative Review. Sports Medicine, 1-20.)

Line 23: Please change to: “154 male and female sprinters”

Line 25: Please change to: “which distinguished elite (PB: 9.95-10.29 s) from sub-elite sprinters”

Line 26: Please change to: “the kinetic demands in the block start”

Line 31: Please change to: “generated considerable amounts”

Line 34: I believe Farris et al. 2016 (Farris, D. J., Lichtwark, G. A., Brown, N. A., & Cresswell, A. G. (2016). The role of human ankle plantar flexor muscle–tendon interaction and architecture in maximal vertical jumping examined in vivo. Journal of Experimental Biology, 219(4), 528-534.) would be an excellent reference to underpin this statement.

Line 41: Please change to: “the jumping height in a squat jump performed without lumbopelvic extensors is approximately 15% lower than when performed with lumbopelvic extensors.” Further, please highlight that these results come from a simulation study.

Line 43: Please change to: “might act as energy generators via”

Line 56: Potentially, you might want to add Funken et al. 2019 (Funken, J., Willwacher, S., Heinrich, K., MüLLER, R. A. L. F., Hobara, H., Grabowski, A. M., & Potthast, W. (2019). Three-Dimensional Takeoff Step Kinetics of Long Jumpers with and without a Transtibial Amputation. Medicine and science in sports and exercise, 51(4), 716-725.) highlighting clear frontal plane contributions to the kinetics of the long jump take-off step.

Line 61: Please change to: “enhancements in training strategies for the track and field block start.”

Line 65: Please change to: “kinetics during the block start.”

Line 86: Please provide the product details for the double sided tape. Potentially, a picture of the experimental setup would be a nice addition to the paper.

Line 86: Please change to: “coordinates attached to the starting blocks”

Line 87: Please change to: “movement of the centre of each block from onset”

Line 115: A reference to the Bezodis et al. (2019) review might make sense here, since it includes a phase definition in line with yours.

Line 120: I think a more detailed description (in addition to figure 1) on how the segment coordinate systems were defined would improve the understanding of the readers. Please provide a detailed description around here.

Line 127 to 136: Could you please provide a more explicit expression on how the point of force application of the ground reaction force was calculated? A formula like:

r = xyz… (where r is the vector to the CoP)

Line 139: If you describe work in this manuscript, do you always reference to net mechanical work? So, is positive work always net positive mechanical work generated during the phase of interest and is negative work always net negative work over the phase of interest? I believe that you could adapt your wording slightly so that it is more clear what you are referring to in the following parts of the manuscript.

Line 147: There are many ways to calculate Cohen’s d or Cohen’s d like parameters, in particular for dependent samples. Therefore, please provide the specific formula you were using to calculate Cohen’s d.

Line 155: I believe it would be good to calculate the normalized horizontal block power (Bezodis, N. E., Salo, A. I., & Trewartha, G. (2010). Choice of sprint start performance measure affects the performance-based ranking within a group of sprinters: which is the most appropriate measure?. Sports Biomechanics, 9(4), 258-269.) for the starts performed in your study. This way, the readers can easiliy perform comparisons with respect to the start performances achieved in your study. You might use leg length in the normalization procedure or if you have not taken this measure use body height instead (see Willwacher, S., Herrmann, V., Heinrich, K., Funken, J., Strutzenberger, G., Goldmann, J. P., ... & Brüggemann, G. P. (2016). Sprint start kinetics of amputee and non-amputee sprinters. PloS one, 11(11), e0166219.).

Line 182 and elsewhere: Is the total generated energy referring to the net work or the sum of absolute positive work and absolute negative work. Please try to clarify this here and elsewhere in the manuscript.

Line 199: I would change this sentence to: “To the knowledge of the authors, this is the first study to describe the kinetic roles of the lumbo-pelvic-hip complex during the track and field sprint start.”

Line 205: Please change to: “generated considerable amounts of kinetic energy (14% of the sum of lumbosacral and lower-limb joint work during the double-stance phase), thus supporting”

Line 207: Please change to: ”in a block start with the body leaning anteriorly,”

Line 209: Please change to: ”except the pelvis in the sagittal”

Line 229: Please change to: ”kinetic energy in the single-stance”

Line 231: Please change to: ”The pelvis is in a considerably anteriorly tilted position”

Line 234: Please change to: ”in an anteriorly tilted position”

Line 239: Please change to: ”important future theme”

Line 243: Please change to: ”abductors exerted mostly positive power, preceded by neglect able negative power generation, suggesting only a small countermovement involved in their action.”

Line 244 and following: If you have determined the normalized horizontal block power parameters for each start, you can perform correlation analyses to see if the amount of work or peak moments are related to start performance. However, I agree that a longitudinal study potentially would be best to strengthen this statement. Still, looking at the correlations would be a first hint towards the actual relationship between start performance and the kinetics of the lumbo-pelvic-hip complex.

Tables and figures

Figure 1: Figure 1c is missing units for the axis descriptions for the graph in the bottom part.

Figure 4: Probably you could change the y-axis description to “Peak Torque [Nm/kg]”

Figure 5: I assume you show net work in this graph. Please also indicate this in the y axis description.

6. PLOS authors have the option to publish the peer review history of their article (what does this mean?). If published, this will include your full peer review and any attached files.

Reviewer #1: No

Reviewer #2: Yes: Steffen Willwacher

---

## [Author Response · Author response to Decision Letter 0]

12 Feb 2020

RESPONSE TO REVIEWER 1

We thank the reviewer for providing useful comments on our paper. We feel that the comments have helped us to improve the quality of our paper. We have revised the manuscript in accordance with your comments. The yellow highlighted parts are the revised points in this version, and the blue highlighted parts are the alternative sentences provided by reviewers.

Our responses to the reviewer’s comments are as shown below: 

COMMENTS:

The research question is new and the experimnts and analysis are very scientific. Also the manuscript is well written. Some minor comments:

We wish to express our appreciation to the reviewers for the comments on our paper.

* page 7 ...prefer to write local university ..not our university

The journal commented that ‘Please amend your current ethics statement to include the ** full name ** of the ethics committee/institutional review board’. Thus, we have modified as:

“The Committee on Ethics of Experimental Research on Human Subjects Graduate School of Arts and Sciences, The University of Tokyo approved the study protocol (reference number: 356-2).” (Lines 84-86)

* page 7 not a go signal ...use e.g. a gun signal

In accordance with the comment, we have revised as:

“a gun-like signal from a starter-signal tool (NT7713C, Nishi, Japan)”. (Lines 89-90)

* page 14: not of 14 % of ..prefer 14 % of...

Reviewer 2 also commented this point. Thus, in accordance with both comments, we have revised it as:

“The lumbosacral extensors generated considerable amounts of kinetic energy (14% of the sum of lumbosacral and lower-limb net joint work during the double-stance phase) ...”. (Lines 254-255)

* page 17 ..in conclusion the first sentence is very long. S Make two short sentences.

In accordance with the comment, we have revised as shown below:

“In conclusion, we found that lumbosacral extension torque was larger than other lumbosacral and lower-limb torques. It was suggested that the lumbopelvic extensors anchor the pelvis by cancelling out both hip extension torques acting on the pelvis, leading to hip extensor-induced thigh sagittal rotations rather than pelvic posterior tilt during a block start.” (Lines 303-307)

///////////////////////////////////////////////////////////////////////////////////

///////////////////////////////////////////////////////////////////////////////////

RESPONSE TO REVIEWER 2

We thank the reviewer for providing useful comments on our paper. We feel that the comments have helped us to improve the quality of our paper. We have revised the manuscript in accordance with your comments. The yellow highlighted parts are the revised points in this version, and the blue highlighted parts are the alternative sentences provided by reviewers.

Our responses to the reviewer’s comments are as shown below: 

COMMENTS:

Overview and general recommendation:

This study presents an analysis of the kinematics and kinetics of the lumbo-pelvic-hip complex and the lower extremity joints during the track and field sprint start. It is the first study that addresses the kinetics of this complex in the literature and therefore adds important knowledge to the field of Sports Biomechanics. The authors need to be commended for their efforts.

The data collection and analysis seems to be well performed. However, the authors should describe certain aspects of their analysis in greater detail. Further, it would be nice to add the calculation of the normalized average horizontal block power values for the analyzed start to allow for a comparison of the start performances in this study to the published literature. The manuscript would also benefit from several improvements in the quality of the English language (see my specific comments).

We wish to express our strong appreciation to the reviewers for their insightful comments on our paper. We feel the comments have helped us significantly improve the paper. We have revised our paper in accordance with each comment as shown below.

Specific comments:

Abstract:

Line 2: Please change to: “joint kinetics during track and field block starts”

In accordance with the comment, we have modified it. (Line 2)

Line 6: Please change to: “mechanical energy during a block start”

In accordance with the comment, we have modified it. (Line 6)

Line 6: Please change to: “start. 3D kinematic and force data”

In accordance with the comment, we have modified it. (Lines 6-7)

Line 7: Please add the performance level (100 m PB times) here.

In accordance with the comment, we have added it. (Line 8)

Line 8: Please change to: “using a motion capture”

In accordance with the comment, we have modified it. (Lines 8-9)

Line 9: Please change to: “kinetics were calculated”

In accordance with the comment, we have modified it. (Line 10)

Line 11: Please change to: “during the block start”

In accordance with the comment, we have modified it. (Line 13)

Line 18-19: Please rephrase this statement. You have not analyzed the effects of strengthening these muscle groups on start performance in this study. Please clearly highlight that this statement is speculative at the moment and that further studies are needed to verify this hypothesis.

In accordance with the comment, we have modified as:

“The results lead to the speculation of the importance of strengthening not only the leg extensors, but also the lumbopelvic extensors, lateral flexors, and hip abductors for block starts. Further training studies to verify this speculation will improve training strategies for the track and field block start performance.” (Lines 19-22)

Introduction:

Line 22: A reference to the recent review of Bezodis et al. might be relevant here, since it provides a good overview on the current sprint start literature and also provides a phase classification for the sprint start. (Bezodis, N. E., Willwacher, S., & Salo, A. I. T. (2019). The Biomechanics of the Track and Field Sprint Start: A Narrative Review. Sports Medicine, 1-20.)

In accordance with the comment, we have cited the reference (Bezodis et al. 2019). (Line 26)

Line 23: Please change to: “154 male and female sprinters”

In accordance with the comment, we have modified it. (Line 27)

Line 25: Please change to: “which distinguished elite (PB: 9.95-10.29 s) from sub-elite sprinters”

In accordance with the comment, we have modified it. (Lines 29-30)

Line 26: Please change to: “the kinetic demands in the block start”

In accordance with the comment, we have modified it. (Line 31)

Line 31: Please change to: “generated considerable amounts”

In accordance with the comment, we have modified it. (Line 36)

Line 34: I believe Farris et al. 2016 (Farris, D. J., Lichtwark, G. A., Brown, N. A., & Cresswell, A. G. (2016). The role of human ankle plantar flexor muscle–tendon interaction and architecture in maximal vertical jumping examined in vivo. Journal of Experimental Biology, 219(4), 528-534.) would be an excellent reference to this statement.

In accordance with the comment, we have added Farris et al. (2016). (Line 39)

Line 41: Please change to: “the jumping height in a squat jump performed without lumbopelvic extensors is approximately 15% lower than when performed with lumbopelvic extensors.” Further, please highlight that these results come from a simulation study.

In accordance with the comment, we have modified it as:

“A simulation study [11] revealed that the sagittal rotations of the thorax and lumbar segment in a vertical jump were generated by the lumbopelvic extensors, not by the hip extensors, and that the jumping height in a squat jump performed without lumbopelvic extensors is approximately 15% lower than when performed with lumbopelvic extensors.” (Lines 44-48).

Line 43: Please change to: “might act as energy generators via”

In accordance with the comment, we have modified it. (Line 49)

Line 56: Potentially, you might want to add Funken et al. 2019 (Funken, J., Willwacher, S., Heinrich, K., MüLLER, R. A. L. F., Hobara, H., Grabowski, A. M., & Potthast, W. (2019). Three-Dimensional Takeoff Step Kinetics of Long Jumpers with and without a Transtibial Amputation. Medicine and science in sports and exercise, 51(4), 716-725.) highlighting clear frontal plane contributions to the kinetics of the long jump take-off step.

In accordance with the comment, we have added the reference (Funken et al., 2019). (Line 63)

Line 61: Please change to: “enhancements in training strategies for the track and field block start.”

In accordance with the comment, we have modified it. (Line 68)

Line 65: Please change to: “kinetics during the block start.”

In accordance with the comment, we have modified it. (Line 71)

Line 86: Please provide the product details for the double sided tape. Potentially, a picture of the experimental setup would be a nice addition to the paper.

In accordance with the comment, we have added as 

“The separated (i.e., not connected by a common bar) starting blocks (NF196BR and NF196BL, Nishi, Japan) were secured onto separate force plates (Fig. 1a) using double-sided tapes (NW-N20, Nichiban, Japan) [19].” (Lines 93-97)

Also, to add the picture of the experimental setup, we have divided Figure as:

“Fig. 1 Experimental setup (a), Marker setup (b), and definitions of lumbar and pelvic segments (c)”

“Fig. 2 Calculation of the centre of pressure (CoP) (a), and a representative example of CoP trajectories (b).”

and added the picture in Figure 1a:

Line 86: Please change to: “coordinates attached to the starting blocks”

In accordance with the comment, we have modified it. (Lines 95-96)

Line 87: Please change to: “movement of the centre of each block from onset”

In accordance with the comment, we have modified it. (Lines 96-97)

Line 115: A reference to the Bezodis et al. (2019) review might make sense here, since it includes a phase definition in line with yours.

In accordance with the comment, we have added the Bezodis et al. (2019). (Line 123)

Line 120: I think a more detailed description (in addition to figure 1) on how the segment coordinate systems were defined would improve the understanding of the readers. Please provide a detailed description around here.

In accordance with the comment, we have modified Figure 1c as shown above and have added the detailed description as:

“For the lumbar SCS definition, the z_lumbar runs from the lumbosacral joint to the thoracolumbar joint, the y_lumbar is the cross product of the z_lumbar and the vector running from the left lower edge of rib to the right lower edge of rib, and the x_lumbar is the cross product of the y_lumbar and the z_lumbar. For the pelvic SCS definition, the x_pelvis runs from the left ASIS to the right ASIS, the z_pelvis is the cross product of the x_pelvis and the vector running from the midpoint of PSISs to that of ASIS, and the y_pelvis is the cross product of the z_pelvis and the x_pelvis.” (Lines 136-143)

Line 127 to 136: Could you please provide a more explicit expression on how the point of force application of the ground reaction force was calculated? A formula like:

r = xyz… (where r is the vector to the CoP)

In accordance with the comment, we have modified the equation as:

“…that was 0 [28] as shown below:

(r^(O→B)+R_(BCS→GCS) 〖r^(B→CoP)〗^' )×f+R_(BCS→GCS) [■(0@0@n_(z_BCS ) )]=n^total (2)

where r^(O→B) is the position vector from the GCS origin to the block coordinate system (BCS) origin expressed in GCS, R_(GCS→BCS) is the transformation matrix from GCS to BCS, r^(B→CoP)' is the position vector from the BCS origin to the CoP expressed in BCS, f is the GRF vector expressed in GCS, [■(0@0@n_(z_BCS ) )] is the free-moment vector applied on the block plane (x_BCS y_BCS plane) expressed in BCS, and n^total is the applied moment vector around the origin of the GCS expressed in the GCS, respectively. From Eqn. 2, the position of the CoP (r^(O→CoP)) can be calculated as:

r^(O→CoP)=r^(O→B)+R_(BCS→GCS) 〖r^(B→CoP)〗^' (3)

〖r^(B→CoP)〗^'=[■(-(n_(y_BCS)^total-[r^(O→B)×f]_(y_BCS ))/f_(z_BCS ) @(n_(x_BCS)^total-[r^(O→B)×f]_(x_BCS ))/f_(z_BCS ) @0)] (4)

where n_(x_BCS)^total n_(y_BCS)^total are the x_BCS and y_BCS components of the n^total expressed in BCS, [r^(O→B)×f]_(x_BCS ) and [r^(O→B)×f]_(y_BCS ) are the x_BCS and y_BCS components of the [r^(O→B)×f] expressed in BCS, and f_(z_BCS ) is the z_BCS component of the GRF vector expressed in BCS, respectively.” (Lines 150-165)

Line 139: If you describe work in this manuscript, do you always reference to net mechanical work? So, is positive work always net positive mechanical work generated during the phase of interest and is negative work always net negative work over the phase of interest? I believe that you could adapt your wording slightly so that it is more clear what you are referring to in the following parts of the manuscript.

We showed only net positive/negative work. 

In accordance with the comment, we have modified “net mechanical work” in the overall manuscript to show that the mechanical work is always net mechanical work.

Line 147: There are many ways to calculate Cohen’s d or Cohen’s d like parameters, in particular for dependent samples. Therefore, please provide the specific formula you were using to calculate Cohen’s d.

In accordance with the comment, we have added the formula as:

“d=|(x_1 ) ®-(x_2 ) ® |/σ_pooled (5)

where (x_1 ) ® and (x_2 ) ® are the mean values and σ_pooled is the pooled SD.” (Lines 179-181)

Line 155: I believe it would be good to calculate the normalized horizontal block power (Bezodis, N. E., Salo, A. I., & Trewartha, G. (2010). Choice of sprint start performance measure affects the performance-based ranking within a group of sprinters: which is the most appropriate measure?. Sports Biomechanics, 9(4), 258-269.) for the starts performed in your study. This way, the readers can easiliy perform comparisons with respect to the start performances achieved in your study. You might use leg length in the normalization procedure or if you have not taken this measure use body height instead (see Willwacher, S., Herrmann, V., Heinrich, K., Funken, J., Strutzenberger, G., Goldmann, J. P., ... & Brüggemann, G. P. (2016). Sprint start kinetics of amputee and non-amputee sprinters. PloS one, 11(11), e0166219.).

In accordance with the comment, we have calculated the normalised average horizontal power (NAHP). 

We have added the calculation method as:

“To quantify the starting performance, the normalised averaged horizontal power (NAHP) [25] was calculated as:

NAHP=(m〖〖 v〗_hori〗^2)/(2 Δt)∙1/(m g^(3/2) l^(1/2) ) (1)

where m is the body mass, 〖 v〗_hori is the horizontal velocity at the end of block phase, Δt is the duration of the block phase, g is the gravitational acceleration, and l is the leg length of the sprinter, respectively.” (Lines 128-134)

And have added the value as:

“The NAHP was 0.55 ± 0.05.” (Line 191)

Line 182 and elsewhere: Is the total generated energy referring to the net work or the sum of absolute positive work and absolute negative work. Please try to clarify this here and elsewhere in the manuscript.

It meant the “net work” not absolute work. In accordance with the comment, we have modified as “net mechanical work” in the overall manuscript to show that the mechanical work is always net mechanical work.

Line 199: I would change this sentence to: “To the knowledge of the authors, this is the first study to describe the kinetic roles of the lumbo-pelvic-hip complex during the track and field sprint start.”

In accordance with the comment, we have replaced it. (Lines 246-248)

Line 205: Please change to: “generated considerable amounts of kinetic energy (14% of the sum of lumbosacral and lower-limb joint work during the double-stance phase), thus supporting”

In accordance with the comment, we have modified it. (Line 253)

Line 207: Please change to: ”in a block start with the body leaning anteriorly,” 

In accordance with the comment, we have modified it. (Lines 257-258)

Line 209: Please change to: ”except the pelvis in the sagittal” 

In accordance with the comment, we have modified it. (Lines 259-260)

Line 229: Please change to: ”kinetic energy in the single-stance” 

In accordance with the comment, we have modified it. (Line 280)

Line 231: Please change to: ”The pelvis is in a considerably anteriorly tilted position” 

In accordance with the comment, we have modified it. (Line 285)

Line 234: Please change to: ”in an anteriorly tilted position” 

In accordance with the comment, we have modified it. (Lines 288-289)

Line 239: Please change to: ”important future theme” 

In accordance with the comment, we have modified it. (Line 293)

Line 243: Please change to: ”abductors exerted mostly positive power, preceded by neglect able negative power generation, suggesting only a small countermovement involved in their action.” 

In accordance with the comment, we have modified it. (Lines 297-299)

Line 244 and following: If you have determined the normalized horizontal block power parameters for each start, you can perform correlation analyses to see if the amount of work or peak moments are related to start performance. However, I agree that a longitudinal study potentially would be best to strengthen this statement. Still, looking at the correlations would be a first hint towards the actual relationship between start performance and the kinetics of the lumbo-pelvic-hip complex. 

In accordance with the comment, we have examined the correlations of them and we have found that the peak lumbosacral extension torque, the net mechanical work by the lumbosacral extensors, and sum of the net mechanical work by the lumbosacral lateral flexors and front hip abductors positively correlated with the sprinting performance.

We have added Methods as:

“To assess the relationships between some lumbo-pelvic-hip kinetic variables and NAHP, we calculated two-tailed Pearson product–moment correlations (r).” (Lines 183-185)

and have added Results as:

“The peak lumbosacral extension torque (r = 0.67, p<0.05, Fig. 7a), the net mechanical work exerted by the lumbosacral extensors during the double-stance phase (r = 0.63, p<0.05, Fig. 7b), and the sum of the net mechanical work exerted by the lumbosacral rear-leg side flexors and front hip abductors during single-leg stance phase (r = 0.68, p<0.05, Fig. 7c) were positively correlated with the NAHP.

” (Lines 229-233)

and have added Figure 7 showing the correlations as:

Fig. 7 Relationships between lumbo-pelvic-hip variables [peak lumbosacral extension torque (a), net mechanical work exerted by lumbosacral extensors during double-stance phase (b), and sum of net mechanical work exerted by lumbosacral rear-leg side flexors and by front hip abductors during single-stance phase (c)] and starting performance (Normalised average horizontal power: NAHP).

Also, we have added Discussion as:

“…thus supporting our hypothesis. Further, we found the positive correlation between the net mechanical work exerted by lumbosacral extensors and the start performance. We speculate that…” (Lines 255-257)

“The peak lumbosacral extension torque was significantly larger than any other lumbosacral and lower-limb torques, and positively correlated with the starting performance.” (Lines 261-263)

and

“The sum of the net mechanical work exerted by the front hip abductors and the lumbosacral rear-leg side flexors was positively correlated with the sprinting performance. The pelvis is in a considerably anteriorly tilted position during the block start. …” (Lines 283-285)

 

Tables and figures

Figure 1: Figure 1c is missing units for the axis descriptions for the graph in the bottom part. 

In accordance with the comment, we have modified the Figure .

 

Figure 4: Probably you could change the y-axis description to “Peak Torque [Nm/kg]” 

In accordance with the comment, we have modified the Figure.

 

Figure 5: I assume you show net work in this graph. Please also indicate this in the y axis description.

In accordance with the comment, we have modified the Figure 6.

---

## [Decision Letter · Decision Letter 1]

24 Feb 2020

Three-dimensional kinetic function of the lumbo-pelvic-hip complex during block start

PONE-D-19-35792R1

Dear Dr. Sado,

We are pleased to inform you that your manuscript has been judged scientifically suitable for publication and will be formally accepted for publication once it complies with all outstanding technical requirements.

With kind regards,

Alena Grabowski

Academic Editor

PLOS ONE

Additional Editor Comments (optional):

Reviewers' comments:

Reviewer's Responses to Questions

**Comments to the Author**

1. If the authors have adequately addressed your comments raised in a previous round of review and you feel that this manuscript is now acceptable for publication, you may indicate that here to bypass the “Comments to the Author” section, enter your conflict of interest statement in the “Confidential to Editor” section, and submit your "Accept" recommendation.

Reviewer #1: (No Response)

Reviewer #2: All comments have been addressed

2. Is the manuscript technically sound, and do the data support the conclusions?

Reviewer #1: (No Response)

Reviewer #2: Yes

3. Has the statistical analysis been performed appropriately and rigorously? 

Reviewer #1: (No Response)

Reviewer #2: Yes

4. Have the authors made all data underlying the findings in their manuscript fully available?

Reviewer #1: (No Response)

Reviewer #2: Yes

5. Is the manuscript presented in an intelligible fashion and written in standard English?

Reviewer #1: (No Response)

Reviewer #2: Yes

6. Review Comments to the Author

Reviewer #1: (No Response)

Reviewer #2: (No Response)

7. PLOS authors have the option to publish the peer review history of their article (what does this mean?). If published, this will include your full peer review and any attached files.

Reviewer #1: No

Reviewer #2: Yes: Steffen Willwacher

---

## [Editor Report · Acceptance letter]

25 Feb 2020

PONE-D-19-35792R1 

Three-dimensional kinetic function of the lumbo-pelvic-hip complex during block start 

Dear Dr. Sado:

I am pleased to inform you that your manuscript has been deemed suitable for publication in PLOS ONE. Congratulations! Your manuscript is now with our production department. 

With kind regards,

on behalf of

Dr. Alena Grabowski 

Academic Editor

PLOS ONE